# Association between Molecular Mechanisms and Tooth Eruption in Children with Obesity

**DOI:** 10.3390/children9081209

**Published:** 2022-08-11

**Authors:** Carla Traver, Lucía Miralles, Jorge Miguel Barcia

**Affiliations:** 1Department of Dentistry, Catholic University of Valencia San Vicente Mártir, 46001 Valencia, Spain; 2Doctoral School, Catholic University of Valencia San Vicente Mártir, 46001 Valencia, Spain; 3Department of Anatomy and Physiology, Catholic University of Valencia San Vicente Mártir, 46001 Valencia, Spain

**Keywords:** pediatric obesity, tooth eruption, dentition, permanent, leptin

## Abstract

Different works have reported earlier permanent teething in obese/overweight children compared to control ones. In contrast, others have reported a delayed permanent teething in undernutrition/underweight children compared to control one. It has been reported that becoming overweight or suffering from obesity can increase gingival pro-inflammatory drive and can affect orthodontic treatment (among other complications). In this sense, little is known about the molecular mechanisms affecting dental eruption timing. Leptin and adiponectin are adipocytokines signaling molecules released in overweight and underweight conditions, respectively. These adipocytokines can modulate osteocyte, odontoblast, and cementoblast activity, even regulating dental lamina initiation. The present review focuses on the molecular approach wherein leptin and adiponectin act as modulators of Runt-related transcription factor 2 (Runx 2) gene regulating dental eruption timing.

## 1. Introduction

According to the World Health Organization (WHO) about 39 million children under the age of five suffer from being obese/overweight. On the other side, undernutrition or malnourishment is present in about 149 million children under the age of five. This paradox represents a significant burden affecting several individual health problems and psychosocial aspects among others.

Focusing on oral related hygiene, one’s nutritional state can affect different dental aspects from caries to malocclusion [1]. Some research works conclude that one’s nutritional state can also modify dental eruption timing. It has been shown that children who are obese and overweight can experience early permanent tooth eruption [2,3,4,5]. In contrast, malnourishment is associated with delayed teething [6,7,8,9]. Apart from genetic and ethnic differences, a lack of essential nutrients and vitamins seems to be directly related to the delayed teething observed in underweight kids. On the other side, the role of leptins as potential causes of early teething in obese kids is part of a weak rationale for these observed phenomena. In spite of the amount of data confirming these two dental alterations, little biological arguments have been singled out as potentially involved mechanisms. Herein, our proposal strongly suggests the involvement of leptin/adiponectin as pivotal elements of this phenomenon.

## 2. Dental Development and Permanent Dentition

Teeth development results from a complicated interaction of the odontogenic epithelium and the ectomesenchyme coming from the neural crest in the jaw/maxilla [10,11]. Initially, primary dentition includes incisors, canines, and molars. These are accompanied by a successional lamina that leads the permanent teeth development. However, the secondary molar dentition is developed by serial addition produced by the extension of the dental lamina in the first molar [12]. Even the dental laminae of permanent teeth can be already found at embryonic stages and can last up to 12 years [13]. During these years the dental successional lamina is normally in a resting state and is activated according to deciduous teeth lost [14]. During this process, some genes and molecules are orchestrated. Among them, Wnt, fibroblast growth factor (FGF) and Hedgehog signaling pathways have been demonstrated to regulate, beyond the bud stage, permanent tooth initiation [15,16,17]. More recently it has been demonstrated that permanent tooth initiation is promoted by mechanical stress release. This mechanical stress inhibits permanent tooth initiation due to the Runt-related transcription factor Wnt (RUNX-Wnt) pathway [18]. Briefly, the pressure exerted by primary teeth activates Runx2, inhibiting successional dental lamina. This blocks permanent tooth initiation. In contrast, the progressive relief of mechanical pressure during deciduous teeth loss decreases Runx2 and increases Wnt expression, leading to permanent tooth initiation.

More recently it has been demonstrated that permanent tooth initiation is promoted by mechanical stress release. This mechanical stress inhibits permanent tooth initiation due to the RUNX-Wnt pathway [18]. Briefly, the biomechanical stress of the primary teeth activates Runx2 inhibiting successional dental lamina. This blocks permanent tooth initiation. In contrast, the progressive relief of mechanical pressure during deciduous teeth loss decreases Runx2 and increases Wnt expression leading to permanent tooth initiation. In line with this, Li et al. [19] pointed out that Runx must be inhibited to promote odontoblast maturation and dentin formation.

## 3. Runx and Energy State

Runt-related transcription factor (RUNX) is a family with three related transcription factors, runt-related transcription factor 1 (RUNX1), RUNX2, and runt-related transcription factor 3 (RUNX3). It has a high conserved sequence of 128 amino acid DNA binding/protein-protein domains, known as the Runt-homology domain [20,21]. RUNX2 determines osteoblast-osteocyte differentiation and regulates chondrocyte division-differentiation during endochondral bone development [22]. The Runx2 pathway is connected to integrin β1 on the cell membrane. Once integrins are stimulated, it promotes extracellular signal-regulated kinase1 (ERK1) activation [23], resulting in Runx 2 transcription and phosphorylation [24]. Fosfatidilinositol 3 kinasa/Akt signalling pathway activation promotes Runt-related transcription factor 2 deoxyribonucleic acid (Runx2-DNA) binding and Runx2 transcription in murine osteoblasts [25].

Although falling outside of the scope of this review, Runx has been widely demonstrated to be related to tumor development, cancer progression, and metastasis in different organs [26,27].

AMP-activated kinase (AMPK) is considered a cellular energy sensor that guides signalling mechanisms leading to homeostatic balance via anabolic or catabolic pathways [28]. RUNX2 is a substrate of AMPK, which directly phosphorylates at serine 118 residue in the DNA-binding domain of RUNX2 [29]. It has been proven that high glucose levels reduced AMPK activity and this fact promotes adipogenesis vs. osteogenesis [30]. Fitting with this, high fat levels also decrease Runx2 expression [31] and even adiponectin increases Runx-2 activity [32]. Adiponectin and leptin are adipocytokines secreted by adipose tissue modulating several functions from cardiovascular modulation to bone metabolism [33,34]. It is generally accepted that obesity is associated to high leptin and low adiponectin levels and inversely for undernutrition [35].

Leptin secretion is increased with feeding and overnutrition when adipocyte number and size is increased [35]. Leptins can affect different pathways, the most relevant of which is the β-oxidation of fatty acids by activating AMP-dependent kinase [36]. It seems that adipose tissue overgrowth results in hypoxia by low vascularization increasing hypoxia inducible factor 1 alpha (HIF1α) and then raising leptin production [37]. However, undernutrition and physical activity both increased adiponectin secretion decreasing adipose tissue volume due to lipolytic activity [38] (Figure 1).

Kapur et al. [39] indicated that leptin receptors (LEPR) negatively modulate bone mechanosensitivity and that genetic variation in LEPR signaling causes a low osteogenic response to loading force. According to Um et al. [40] leptins can promote cementoblast/odontoblast differentiation in dental mesenchymal cells. Periodontal ligament fibroblasts over expressed pro-inflammatory factors such as the receptor activator of Nf-Kappa b (RANKL) in the presence of leptin under mechanical strain [41]. Leptin levels can be found and detected in gingival crevicular grooves in healthy subjects compared to those with periodontal disease [42]. Even more, during orthodontic treatment, leptin levels are increased one day after intervention and decreased one week after [43]. Despite the fact that most clinicians suggest a relationship between obesity and tooth movement, after reviewing several studies, it cannot be confirmed that obesity affects tooth movement [44]. Saloom et al. [45] published a prospective clinical cohort study with 55 teenagers (obese vs. normal-weight) and observed significant higher tooth movement in the obese group. Even more, they also found significantly higher levels of leptin and RANKL in this group. These findings support again the potential role of leptins on periodontal and dental evolution and revive the discussion of obesity and its role in orthodontic movement.

Secreted by fat cells and salivary gland epithelial cells [46]. Adiponectin can be bound to adiponectin receptor 1 (AdipoR1) and adiponectin receptor 2 (AdipoR2) receptor types activating the adenosine monophosphate-activated protein kinase pathway (among others) [47,48]. Adiponectin increases fatty acid oxidation, glucose uptake, insulin sensitivity, and can also present anti-inflammatory effects [34,49,50]. Both adiponectin receptors are found in periodontal ligament fibroblasts and osteoblasts [51,52]. According to Marjan Nokhbehsaim et al. [53] adiponectin promotes beneficial effects on periodontal ligament cells by increasing growth factor production and self-promoting adiponectin production. It has been shown that high adiponectin levels increase Runx-2 in cementoblasts, as well as promoting osteoblast differentiation and migration [32,54,55,56]. In this work, Yong et al. [32] also reported that high adiponectin level exposure increases alkaline phosphatase, osteocalcin, bone sialoprotein, osteocalcin and osteoprotegerin nucleic messenger (mRNA) levels. In contrast, the use of physiological adiponectin concentrations did not result in as significant a response. This means that adiponectin potentially modulates many periodontal-related factors (albeit without playing an exclusive role).

On the other hand, adiponectin decreases pro-inflammatory factors (e.g., tumor necrosis factor alpha). In this sense, Kraus et al. [57] pointed out that low levels of adiponectin in obese/overweight individuals could be related to periodontal inflammation and destruction.

Adiponectin is also involved in cell homeostasis by regulating the mitogen activated protein kinase pathway (MAPK) [58]. Luo et al. [59] suggest that adiponectin receptor-JNK pathway regulates osteoblast proliferation and that adiponectin receptor-P38 modulates differentiation. Previously, Kadowaki et al. [60] indicated that adiponectin stimulated osteogenesis involving adiponectin receptor 2 (P38-AdipoR1) and by increasing Runx-2.

In the literature we also find several environmental factors that affect the tooth eruption. As is well known, tooth eruption is a long lasting process (it often lasts years) and there are many factors that can modify this process (Figure 2). One can see how an overweight child or an obese child may have an advanced tooth eruption process [3,4,5,8,61,62,63,64,65].

## 4. Conclusions

In spite of the amount of data indicating that being obese/overweight can promote early permanent tooth eruption, on the contrary undernutrition leads to delayed permanent teething. There is a considerable lack of knowledge and rationale about the molecular signals giving response to these phenomena.

This review investigates the potential relation of leptin and adiponectin as molecular modulators of dental development. It has been demonstrated that leptins are present in the crevicular fluid of healthy subjects and that leptin levels can be altered during orthodontic mechanical strain. Adiponectin and leptin can also promote osteoblast and odontoblast differentiation.

We consider Runx 2 as a potential regulator on this phenomenon since it is a substrate of AMPK. AMPK works as energy sensor and regulates Runx 2. Runx 2 acts as a bone and dental development regulator [18,19]. Both adiponectin and leptin can also affect Runx 2 activity, and this has been proposed as an approach to explain early or delayed dental eruption for both overweight and underweight children.

## Figures and Tables

**Figure 1 children-09-01209-f001:**
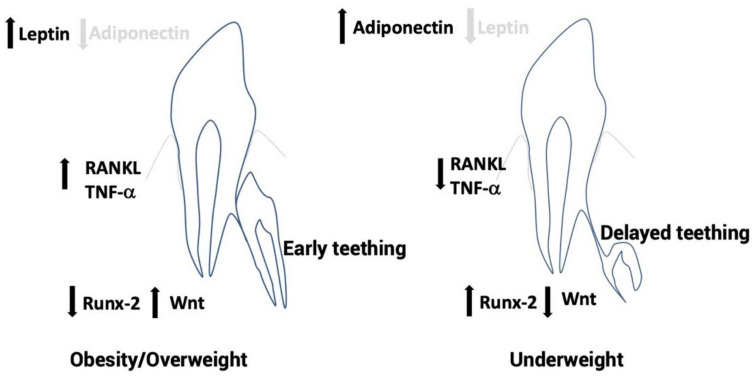
Under obesity/overweight conditions, early permanent teeth eruption would be explained by increased leptin levels that decrease runt-related transcription factor 2 (Runx 2) expression increasing Wnt gene expression, receptor activator of NF kappa-b (RANK-L), and TNF-α. In underweight conditions, delayed permanent tooth eruption would be explained by increased adiponectin levels or decreased adiponectin. This promotes Runx 2 expression resulting in Wnt reduced expression and decreased pro-inflammatory RANK-L and TNF-α.

**Figure 2 children-09-01209-f002:**
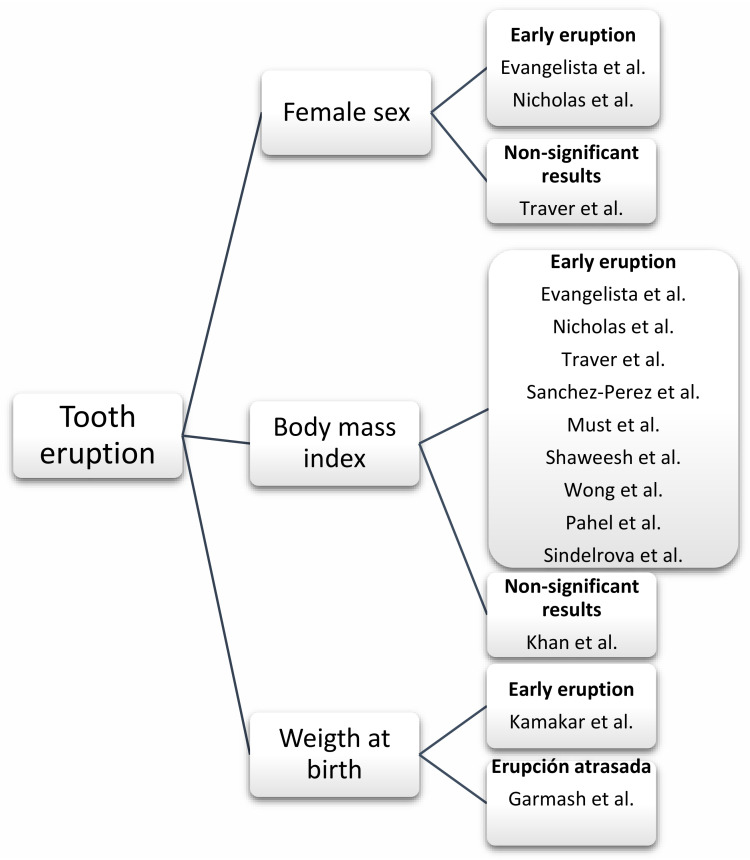
Relationship between obesity and non-molecular factors [3,4,5,8,61,62,63,64,65,66,67,68].

## Data Availability

Not applicable.

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
