# Peer review of "Association between Molecular Mechanisms and Tooth Eruption in Children with Obesity"

_children, 2022, doi:10.3390/children9081209_

Round 1

Reviewer 1 Report

This review manuscript revised the potential relation of leptin and adiponectin as molecular modulators that promote osteoblast and odontoblast differentiation. These modulators seem to influence dental eruption. The manuscript contains five keywords, one figure, and sixty-two references. Overall, it is a correct manuscript, although some remarks are made.

Specific comments on different sections of the manuscript are exposed.

Title
The title is not very informative about the content of the article. Nothing is mentioned about obesity or undernutrition, but instead, it focuses on leptin and adiponectin and eruption timing.

Keywords
The manuscript presents five keywords. For keywords, where possible, please use Medical Subject Headings terms (MeSH Terms). All of them are MeSH terms, except “Runx 2 gene”. An alternative MeSH term proposed could be “dentition, permanent” better than “permanent dentition”. Nevertheless, these suggestions about keywords are optional, not mandatory.

General comments
Page 2, line 50. Abbreviations and acronyms should be explained the first time they are used, e.g. Wnt, FGF. Please do the same with the rest of the abbreviations not explained in the text.

Page 2, line 64. To make text understanding easier, if the author's name appears in the text, place the reference number immediately after the name, not at the end of the sentence or paragraph.

Page 4, line 126. In the reference list, number 54 indicates the first author's full name (Marjan Nokhbehsaim), not just his last name (Nokhbehsaim). Please correct this in the text.

References
Total number of manuscript references: 62.
This is the weakest section of the manuscript. It requires a comprehensive revision. There are several different reference formats. None of them match to reference format suggested by the journal. References should be checked carefully to transcribe them accurately.

Please, consult the instructions for authors and the reference examples at the following link: https://www.mdpi.com/journal/children/instructions

 Figures
Total number of the manuscript figures: 1.
The figure has an appropriate figure legend.

Author Response

I have modified everything indicated and added in the article. I attach it below.

Reviewer 2 Report

"the pressure exerted by primary teeth activates Runx2 inhibiting succes-sional dental lamina.  This fact....".  How can this be if the primary teeth are superior to the permanent teeth in the mandible and not pressing on them.    Is it a fact or a conjecture?  Also "obesity is associated to high leptin"  - please provide a reference that obesity results in high leptin levels.

"adipose tissue overgrowth results on hypoxia by low vascularization increasing hypoxia inducible factor 1 alpha (HIF1) and then raising leptin production"  what is the evidence this occurs locally in the jaw?  

Tooth eruption is a highly time variable phenomenon.  Provide references that this is accelerated in obese children.  There are a number of factors discussed  and it would be easier to follow if you provided a diagram showing the interaction of the different factors

Author Response

 "Consulte el archivo adjunto."
